# Safety Attitude as a Predictor of the Sense of Threat in the Workplace, Using the Example of Airport Ground Staff

**Małgorzata Dobrowolska [1,\*], Marta Stasiła-Sieradzka [2] and Jarosław Kozuba [3]**

[1] Institute of Education and Communication Research, Silesian University of Technology, 44-100 Gliwice, Poland

[2] Faculty of Social Science, Institute of Psychology, University of Silesia, 40-007 Katowice, Poland; marta@sieradzki.pl

[3] Faculty of Transport and Aviation Engineering, Silesian University of Technology, 44-100 Gliwice, Poland; jaroslaw.kozuba@polsl.pl

[\*] Correspondence: malgorzata.dobrowolska@polsl.pl

**Abstract:** In its development, air transport is obliged to take into account guidelines related to its sustainability. An important element thereof consists of making sure that the working conditions for all the workers employed in this sector are safe and healthy. The aim of this research was to analyze the relationship between the attitude towards safety and the perceived feeling of threat in the workplace among respondents belonging to the airport ground staff occupational group. The research verified the following: 1. whether a relationship exists between the safety attitude (in the following dimensions: affective, cognitive, and behavioral) and the feeling of threat in the workplace (in the following dimensions: internal discomfort related to the fear of potential threats, the fear of current threats, and seeking to avoid threats); 2. whether attitudes towards safety act as predictors of the feeling of threat in the workplace. A total of 299 individuals took part in the research. Purposive sampling was used, based on criteria related to the respondents being members of airport ground staff. The Safety Attitudes Questionnaire by M. Znajmiecka-Sikora was used in the research. The feeling of threat was assessed using Feeling of Threat at Work Questionnaire by Mamcarz. The results obtained confirmed the assumed negative correlation between the main variables, and at the same time they revealed that among the studied safety attitude components, only the affective component makes it possible to predict the general feeling of threat, being its negative predictor ($\beta = -0.14$; *s.e.* $= 0.33$; $t = -2.27$; $p = 0.024$). The affective component of the safety attitude also constituted a negative predictor of one of the dimensions of the studied feeling of threat, namely seeking to avoid threats ($\beta = -0.22$; *s.e.* $= 0.17$; $t = -3.66$; $p < 0.001$).

**Keywords:** attitudes towards safety; attitudes towards occupational safety; airport ground staff

## 1. Introduction

Dynamically developing air transport determines efficient economic exchange, effective movement of capital, goods, services, and labor, as well as free movement of people. It is an extremely innovative branch in organizational, technical, and technological terms, and it also has an impact on the environment. In its development, air transport is obliged to take into account guidelines related to its sustainability in terms of economic, environmental, and social goals. The most important social goals from the point of view of work and organizational psychology concern the sphere of human labor and are defined as particular care for equality in employment, a corporate culture that guarantees human well-being, as well as safe and healthy working conditions.

The issues of sustainable development of air transport concerning its social dimension relate to caring for the work safety of the people employed in it and concern to an equal extent manufacturers of aviation equipment, airport workers, aircraft personnel, and entities operating in the market environment.

When describing occupational work, as one of the basic forms of human activity, researchers consider it both from the point of view of its relationship with the individual's well-being and with the threat felt by the individual [1–3]. The emerging inappropriate bidirectional relationships between superiors and subordinates, mobbing, aggression, pathology of systemic management solutions in the organization, including its relations with the external environment, the introduction of areas of work with increasing cognitive complexity, as well as the need to constantly learn and adapt to changes are only selected pieces in this kaleidoscope. Each occupational group carries out its tasks in a specific work environment, and the threats occurring therein are often specific and unique in their nature.

Together with information technology, air transport development is referred to as the foundation and pillar of globalization infrastructure. Its development is becoming a challenge in terms of ensuring an appropriate level of work safety both for the people employed in this sector and for all users [4].

As emphasized by M. Zieliński [5], responsibility for safety in air transport should extend to all individuals connected with aviation operations in any way, i.e., both flight crew and ground staff—i.e., the support personnel. In the broadest sense of the term, this refers to all those who are involved in design, manufacture, technical operation and maintenance, flight operations, and air traffic control.

Airport ground staff members include ground administration and supervision, passenger service, baggage, cargo and mail handling, apron handling of aircraft (refueling and maintenance), management of air operations, and administrative matters related to the crew, ground transport, and provision of food and beverages to passengers (catering).

The working environment of these staff is burdened with numerous risks related to the operation of airports. Natural (objective) threats include weather anomalies and other natural factors preventing airport use (strong wind, heavy rainfall, lightning, heavy snowfall and blizzards, fog, or airborne volcanic ash (which the world experienced in April 2010)). Threats caused by humans, on the other hand, include planting or the threat of planting of explosive materials and devices in airport structures and facilities, and disturbance of public order. Another threat type occurring in the working environment of this occupational group involves fires of structures or aircraft, technical failures of various types and epidemiological threats (particularly evident during the COVID-19 pandemic), terrorist threats (seen to intensify after 11 September 2001), and the associated states of increased vigilance. Other threats which are not without importance are associated with the use of transport equipment, as well as working around moving machine parts in confined, poorly-lit spaces.

According to T. Tomaszewski [6], a threat is a situation in which there is an increased likelihood of losing values humans cherish, such as life, health, property, rights, social position, reputation, outputs of one's own work, well-being, and self-esteem. An individual fearing a potential loss is more deeply affected by the situation they find themselves in. This also stems from the subjective feeling of lack of control over environmental factors (e.g., weather conditions hindering the operation of airports) and from predictions and concerns about the possibility of a threat situation occurring in the workplace, or its imagined consequences (threat related to potential contact with someone suffering from a contagious disease).

As P. Mamcarz [7,8] emphasizes, multiple cognitive systems contribute to the development of the feeling of threat, perceived as a mechanism. It is understood as experiencing concerns about the effects of existing or potential threats. It involves both an emotional aspect (experiencing negative emotions) and a cognitive one (content of the pictured threat, attribution, retrospection). The whole process starts with the individual's participation in the specific situation. The individual searches their environment to obtain information about the situation they have found themselves in, and selects the available data. The result of this assessment may be positive or negative. In either case, the individual imagines the potential scenarios for its further development. According to what they imagine, they take specific

actions aimed at implementing the optimal action plan which is most effective in the given situation, supposed to avoid or minimize the threat consequences in the case of a negative assessment.

The feeling of threat may be analyzed using three dimensions. The first is the internal discomfort associated with the anticipation of the threats possible in the given working environment, the concern about potential hazardous situations (e.g., fire, viral infection, fall from a height) that cause internal anxiety, and fear of the consequences of such hazardous events. Another dimension involves the concern about existing threats, which really occur in everyday working conditions (e.g., constantly working in the vicinity of moving equipment in operation, working in a magnetic field, or in dusty spaces), related in turn to the feeling of anxiety resulting from the fact of performing work in a dangerous environment threatening the health and life of the worker. The final dimension involves seeking to avoid the threats, and includes actions at the behavioral level, associated with control activities and with refraining from risky actions during work, and cognitive processes such as intensified observation and analysis of the work situation (e.g., the need for continuous, increased attention faced with the physical threats present in the working environment, increased attention when performing work in which omissions and mistakes may result in creating a situation of threat to oneself or others—the work of pilots, aircraft mechanics, medical laboratory workers) [6].

The perception of threats is influenced by cognitive structures: knowledge, patterns, judgments, and beliefs. Memory resources activated in the situation of searching the environment and cognitive representations of the recognized objects seem equally important [9]. Attitudes towards safety are also extremely important in the social and professional functioning of individuals. They have a substantial influence on the way information is processed—not only facilitating its reception, but also deforming it. In addition, they are inextricably linked with the individual's eventual behavior [10].

Studies focusing on attitudes most often adopt a three-factor definition of the latter, assuming the existence of a cognitive component, an emotional-evaluative (affective) component, and a behavioral component. The cognitive component involves the resource of human knowledge and the resulting beliefs about the object of the attitude. The affective component involves the positive or negative feelings towards the object of the attitude. The behavioral component involves the consequences (effect) of the cognitive and affective components. Thus, the individual's knowledge, their beliefs, and emotional attitude induce them to manifest certain behavior in a situation related to the occurrence of an event representing the specific object of the attitude, which may stem from a threatening factor present in the work environment [11].

The worker's attitude towards work safety may be defined as the general set of relatively permanent dispositions to perceive and judge the principles of work safety, to react emotionally to them, and to perform work safely [12,13]. The cognitive component refers to knowledge about the principles and ways of performing work safely, the risks occurring in work processes, and near misses. This knowledge is passed on during occupational safety training and job-specific instruction training, as well as acquired through experience. The affective component manifests itself in an emotional attitude towards observing safety rules, and contains positive or negative feelings towards the object of the attitude. In the case of work safety, this may be the worker's emotional attitude towards the observance of occupational health and safety rules in the organization, including the principles on the basis of which the employer notifies the workers about work-related risk, about the involvement of the relevant services and superiors in the processes of minimizing risks, and about the ways of providing protection against their consequences. The behavioral component, in turn, refers to displaying certain behaviors at work, contributing to safety [14,15]. Therefore, the worker's knowledge about threats and about personal protection methods, as well as a positive attitude towards the ways of implementing this protection suggested by the employer, may result in safer behavior on the worker's part, in accordance with the procedures in force, and reduce the feelings of threat in the workplace which the worker might have.

Research carried out so far has focused on analyzing the relationship between the worker's attitude towards safety and person-related traits (age, education) [14,16], as well as organizational

characteristics (working conditions—the plant and machinery used and its technical condition, the size of the company [14], work organization, such as—information flow, time pressure [17]), company size [4,18,19], organizational and safety climate [20,21], relations with superiors [14] and co-workers [22], and distance towards the threat itself [23]. It has also been noted that the attitude towards safety is negatively correlated with the tendency to perceive and take both instrumental and stimulation risks [10].

Following a review of the references, it was assumed that the research presented here would be aimed at supplementing the existing knowledge by studying the relationship between safety attitudes and the feeling of threat in the workplace on the group chef airport ground staff members. Recognizing the above relationship creates an area for psychologists' activities both in the field of basic research and in that of applied research, by undertaking various activities (e.g., social campaigns, promotion of safe behaviors in the organization as part of incentive systems, and other HR tools) to promote modification and consolidation of pro-safety attitudes.

## 2. Research Objectives

The aim of this research was to study the relationship between the attitude towards safety and the perceived feeling of threat in the workplace among respondents belonging to the airport ground staff occupational group. The following research questions were formulated following a review of the references:

1. Is there a relationship between the safety attitude (including all its studied dimensions, i.e., affective, cognitive, and behavioral) and the feeling of threat in the workplace (including all its studied dimensions, i.e., internal discomfort related to the fear of potential threats, the fear of existing threats, and seeking to avoid threats)?

2. Does the safety attitude (explanatory variable), including all of its studied dimensions, i.e., affective, cognitive, and behavioral, act as a predictor of the feeling of threat in the workplace (explained variable), including all of its studied dimensions, i.e., internal discomfort related to the fear of potential threats, the fear of existing threats, and seeking to avoid threats)?

The following research hypotheses were put forward:

**Hypothesis (H1).** *There is a relationship between the safety attitude (including all its studied dimensions, i.e., affective, cognitive, and behavioral) and the feeling of threat in the workplace (including all its studied dimensions, i.e., internal discomfort related to the fear of potential threats, the fear of existing threats, and seeking to avoid threats). The relationship is assumed to be of a negative nature; the more positive the safety attitudes, the lower the feeling of threat in the workplace.*

**Hypothesis (H2).** *The safety attitude (including all its studied dimensions, i.e., affective, cognitive, and behavioral) acts as predictor of the feeling of threat in the workplace (including all its studied dimensions, i.e., internal discomfort related to the fear of potential threats, the fear of existing threats, and seeking to avoid threats).*

## 3. Materials and Methods

The research was conducted in the territory of Poland in the period from June 2019 to February 2020. It was of a quantitative nature and it was carried out using questionnaires, which the respondents filled out using the paper-and-pencil method with consent given by the authorized representatives of the respective working establishments run by the organizations where the respondents were employed. The research was conducted in compliance with the ethical standards in line with the provisions of the Declaration of Helsinki. All the respondents gave their voluntary and informed consent to their participation in the study. The results obtained were coded, with the anonymity of the participants ensured.

A total of 299 individuals took part in the research, accounting for 91% of the people provided with the questionnaires. Purposive sampling was used, based on criteria related to the respondents being

members of airport ground staff. They included ground handling staff and cabin crew with regard to supplying fuel, lubricants, and other technical materials to aircraft, technical-administrative staff providing services to aircraft and crews, and flight operations staff. The average age of the respondents was 46.8; women constituted 14.6% and men 85.4% of the respondents.

The Safety Attitudes Questionnaire by M. Znajmiecka-Sikora [10] was used in the research. Following the assumed theoretical concept, safety attitude is perceived as the general set of relatively permanent dispositions to perceive and judge the principles of work safety, to react emotionally to them, and to perform work in a safe manner. When structuring the questionnaire, it was assumed that attitudes had a strong influence on behaviors—in this case, on behavior in line with safety rules or on risky and dangerous behavior. The tool consists of 18 statements covering 3 attitude aspects:

- Cognitive component—concerning opinions and beliefs, shaped by knowledge and experience. This component is also connected with mental processes, e.g., perception and judgment, taking place during the observation of various events. Example item: I am capable of judging and distinguishing individuals engaging in risky behavior, potentially exposing their own or other people's lives or health to a threat;
- Affective component—referring to the positive or negative emotions associated with certain situations as well as their emotional evaluation. Example item: Whenever I participate in dangerous and risky situations, I feel discomfort and negative emotions (e.g., stress, panic, nervousness);
- Behavioral component—referring to specific behaviors and actions. It is connected with effective actions into which the results obtained in the other two components may be translated. Example item: Unless I have sufficient knowledge, experience or competence, I refrain from performing tasks that may be dangerous and risky.

The respondents took a position on each of the statements using a five-point scale: DA (definitely agree), Y (yes), HS (hard to say), N (no), DD (definitely disagree). Each of the three attitude aspects is described by six items; score of min 1—max 30; the higher the score, the more positive the attitude towards the specific attitude aspect. The reliability of the questionnaire was calculated using Cronbach's alpha, which for the overall result was 0.849, and for the individual dimensions it ranged from 0.750 to 0.690.

The sense of threat was assessed using the Feeling of Threat at Work Questionnaire by Mamcarz [8]. The questionnaire is used to measure the feeling of threat, understood as experiencing fears about the effects of existing or potential threats in the workplace. These fears are based on previous negative experiences, and the state described by the respondents is accompanied by an internal sense of discomfort.

The scale is used to study the general degree of the feeling of threat and its three dimensions:

- Internal discomfort related to the fear of potential threats—referring to feelings resulting from participation in threats. Example item: When working, I feel inner anxiety;
- Fears of existing threats—referring to work in an environment judged as threatening. Example item: During work, I think that something bad might happen;
- Seeking to avoid threats—referring to actions supposed to reduce the impact of threats. Example item: I analyze again situations that occurred during the working day.

The questionnaire contains 54 statements. The respondents take a position on them using a Likert scale, with 1 meaning "never" and 5 meaning "very often". Internal discomfort related to the fear of potential threats is described by 36 items, score of min 1—max 180; fears of existing threats described by 12 items, score of min 1—max 60; seeking to avoid threats described by 6 items, score of min 1—max 30; the higher the score, the higher the sense of threat. The reliability of the questionnaire was calculated using Cronbach's alpha, which for the individual dimensions ranged from 0.964 to 0.688.

The study results were analyzed using the JASP 0.11.1 package (JASP Team, 2019).

## 4. Results

Table 1 presents the descriptive statistics of the studied variables and Cronbach's alpha internal consistency coefficients referring to the reliability of all the scales used in this study.

**Table 1.** Reliability levels and descriptive statistics for the studied variables.

|  |  | Cronbach's Alpha | *M* | *Me* | *SD* | Skewness | Shapiro-Wilk *W* |
|---|---|---|---|---|---|---|---|
| | Cognitive component | 0.77 | 16.55 | 17.00 | 3.67 | 0.40 | 0.98 *** |
| Safety | Affective component | 0.78 | 18.16 | 18.00 | 4.54 | −0.25 | 0.97 *** |
| Attitude (SA) | Behavioral component | 0.61 | 17.21 | 17.00 | 2.78 | −0.10 | 0.98 ** |
| | General attitude | 0.80 | 51.92 | 53.00 | 8.38 | −0.54 | 0.95 *** |
| | Fear of potential threats | 0.94 | 73.53 | 71.00 | 18.05 | 0.83 | 0.92 *** |
| Feeling of | Fear of existing threats | 0.84 | 23.14 | 22.00 | 6.10 | 1.15 | 0.92 *** |
| threat (FTWQ) | Seeking to avoid threats | 0.68 | 13.49 | 13.00 | 3.46 | 0.62 | 0.96 *** |
| | General feeling of threat | 0.95 | 110.16 | 105.00 | 24.94 | 1.07 | 0.92 *** |

Note: ** $p < 0.01$; *** $p < 0.001$.

The analysis indicated a satisfactory level of reliability for most of the scales used, with the lower boundary (>0.60) reliability level for the scale studying the behavioral component of the safety attitude and seeking to avoid threats, understood as an element of the sense feeling of threat.

The assumption of normal distribution of the variables was tested using the Shapiro-Wilk W-test. It was demonstrated that the results for all the variables deviated in terms of their shape from the normal distribution, but the level was low (below ±2.0). Consequently, Pearson's r linear correlation coefficient was used as a measure resistant to a slight deviation from normality in a relatively large sample (N > 20) [24].

The correlation analysis results are presented in Table 2.

**Table 2.** Relationships between safety attitude and sense of threat—correlation analysis.

|  | CC | AC | BC | SA | FT 1 | FT 2 | FT 3 | FT |
|---|---|---|---|---|---|---|---|---|
| FT | - | | | | | | | |
| AC | 0.27 *** | - | | | | | | |
| BC | 0.58 *** | 0.31 *** | - | | | | | |
| SA | 0.77 *** | 0.76 *** | 0.75 *** | - | | | | |
| FT 1 | −0.13 * | −0.15 * | −0.10 | −0.17 ** | - | | | |
| FT 2 | 0.03 | −0.07 | 0.04 | −0.01 | 0.72 *** | - | | |
| FT 3 | 0.07 | −0.17 ** | 0.09 | −0.03 | 0.54 *** | 0.50 *** | - | |
| FT | −0.08 | −0.15 * | −0.05 | −0.13 * | 0.98 *** | 0.84 *** | 0.65 *** | - |

**Notes:** CC—Cognitive component, AC—Effective component, BC—Behavioral component, SA—General safety attitude, FT 1—Fear of potential threats, FT 2—Fear of existing threats, FZ 3—Seeking to avoid threats, FT—General feeling of threat. * $p < 0.05$; ** $p < 0.01$; *** $p < 0.001$.

The results of the analysis indicated the existence of moderate intercorrelations between the test subscales, as well as strong correlations between subcomponents and overall results in the test, which confirms the coherent structure of the constructs.

With respect to the H1 assumptions, it can be observed that the general attitude towards safety displayed a weak negative correlation with the general feeling of threat (r = 0.13; $p = 0.023$). With respect to its assumptions concerning the studied correlations between the dimensions of the main variables, the analysis indicated a weak negative correlation between the cognitive component of the safety attitude and the fear of potential threats (r = −0.13; $p = 0.021$). The affective component of this attitude also displayed a weak negative correlation with the discomfort associated with fears of potential threats (r = −0.15; $p = 0.012$), as well as with seeking to avoid threats (r = −0.17; $p = 0.004$) and the general feeling of threat (r = −0.15; $p = 0.012$). In addition, the overall safety attitude displayed a weak negative correlation with the fear of potential threats (r = −0.17; $p = 0.003$). No statistically

significant relationships were found between the behavioral component and feeling of threat and its dimensions.

Subsequently, a linear regression analysis was performed to check whether individual elements of the safety attitude could constitute predictors of the feeling of threat. The analysis involved the creation of four regression models, in which the explained variables were the overall feeling of threat (model 1) and its elements (models 2, 3, and 4). The results of the analysis are presented in Table 3.

**Table 3.** Safety attitude as a predictor of the sense of threat—regression analysis results.

|  | Model 1 General Feeling of Threat | Model 2 Fear of Potential Threats | Model 3 Fear of Existing Threats | Model 4 Seeking to Avoid Threats |
|---|---|---|---|---|
| Cognitive component | −0.06 | −0.10 | 0.02 | 0.06 |
| Affective component | −0.14 * | −0.12 | −0.09 | −0.22 *** |
| Behavioral component | 0.02 | −0.01 | 0.05 | 0.12 |
| $R^2$ | 0.01 | 0.02 | 0.01 | 0.04 |
| F | 2.36 | 3.16 * | 0.93 | 5.33 ** |

Note: The table shows the values of standardized regression coefficients. The adjusted $R^2$ value was indicated as the coefficient of determination. * $p < 0.05$; ** $p < 0.01$; *** $p < 0.001$.

The regression analysis shows that among the components of the safety attitude, only the affective component makes it possible to predict the general feeling of threat, being a negative predictor (β = −0.14; s.e.). = 0.33; t = −2.27; $p = 0.024$). The other components did not make it possible to predict the general feeling of threat. The constructed model demonstrated a poor fit to the data (F [3.295] = 2.36; $p = 0.072$) and explained only 1% of the variance with regard to the feeling of threat.

As far as the subcomponents of the feeling of threat are concerned, the regression analysis indicated that none of the components of the safety attitude made it possible to predict fears of potential or existing threats. The affective component of the safety attitude, in turn, was a negative predictor of seeking to avoid threats (β = −0.22; s.e. = 0.17; t = −3.66; $p < 0.001$). The constructed model demonstrated a good fit to the data (F [3.295] = 5.33; $p = 0.001$) and explained 4% of the variance with regard to the feeling of threat.

## 5. Discussion

The results obtained confirmed the assumed negative correlation between the main variables, and at the same time they revealed that among the studied safety attitude components, only the affective component makes it possible to predict the general feeling of threat, being a negative predictor. The affective component of the safety attitude also constituted a negative predictor of one of the dimensions of the studied feeling of threat, namely seeking to avoid threats.

The research results presented here, although limited only to a selected occupational group, encourage reflection on actions aimed at minimizing the feeling of threat by building a climate of trust, resulting in positive emotions towards the employer's actions in favor of worker safety. The predominance of positive or negative attitudes towards safety in the working environment is fundamentally determined by the employees' belief in the value attributed to safety at work by their employers [12]. The feeling of safety as a certain kind of positive attitude towards the acceptance of risk is of particular importance in the area of work performed in conditions of numerous threats [25]. Individuals working in dangerous environments trust to a significant extent that their employer will make every effort to make sure that their working conditions are as safe as possible, and that the employer will make appropriate measures available and take appropriate action in the event of a threat.

Attitudes towards safety at work are the result of the impact of many different factors, but the important thing is that, according to the opinions of social psychologists, they can be learned, as well as unlearned or changed, even though the process is very difficult [14,15]. Some attitudes are created by copying from other people considered as role models. They can also be shaped as a result of testing (trying out) certain behaviors and obtaining rewards or punishment for them. Individuals

may be persuaded or encouraged to display certain attitudes, e.g., under the influence of persuasive messages. However, the emotional-affective and at the same time evaluative attitude towards safety constitutes an internal construct built on the basis of the individual's experiences. As researchers studying these aspects emphasize [14,15,25], it is based on a certain kind of trust towards the employer regarding all the forms of the latter's impact on the work environment, which through them becomes less threatening for the worker.

In this day and age of increasingly rapid technological progress in almost every sphere of human life, better and better technical solutions and procedures related to extensive prevention of various threats can be noticed. However, as numerous researchers have already emphasized [26–28], limiting activities to this area alone is not effective. Research on the role of leaders in the creation of positive attitudes towards safety already refers to the conclusions by K. Lewin [29], namely that at the group level, a process of social learning takes place through multiple observations of the superior's behaviors and of the actions of individual team members which the superior rewards. Moreover, the behavior displayed by leaders creates an area for interpretation of priorities across the organization—including those related to work safety and to the attitude towards the threats occurring in the working environment. Flin and Yule [30] emphasize that managers on all levels should clearly demonstrate their positive attitudes towards safety in all the dimensions—affective, cognitive, and behavioral. In fact, appropriate shaping of the attitudes of subordinate employees is necessary for the shaping of a safe working environment. The diffusion of a safety attitude within the organization is based on the observed behaviors of the management. The main factors listed as ones influencing this diffusion are the following: the trust which the workers have in their superiors, their dedication to the shaping of safe working conditions, the displayed attitude to threats, supportive supervision, looking after the group's cohesion, consent to the participation of staff in pro-safety activities. All these factors carry an emotional charge which seems to come to the foreground as a priority in terms of reducing the feeling of threat among individuals working in high-risk conditions [31].

Zohar and Luria [32] pointed out that at the practical level, contributing to the shaping of positive attitudes towards safety and reducing the feeling of threat consists not only of applying the procedures and using the protective equipment required by the legislative provisions. An important element here is precisely a positive emotional attitude towards these actions on the part of the staff. This attitude is in turn built on trust, developed in situations when people see that they are being looked after in a genuine way, for example, through purchases of expensive, but also the best available protective equipment, by organizing work and defining performance standards in such a way as to make it possible to comply with occupational health and safety rules, the absolute requirement to provide help in the event of a threat by initiating efficient rescue operations, not concealing information about active threats and their implications for human health and life. The described mechanism, based on a positive emotional dimension of the safety attitude displayed by staff, leading to a reduced feeling of threat, is extremely fragile. It is particularly vulnerable in critical situations, when any omissions or negligence when it comes to the containment of the effects of threats are exposed. It is extremely difficult to restore a positive safety attitude after such a situation at the level of each of its components, but the largest difficulty seems to be related to the affective component here, as a significant predictor of the feeling of threat.

Research findings in the field of social sciences concerning the development of attitudes towards safety and their relationship to the sense of risk in an occupational situation and the role of managerial staff in this process are consistent with the safety management policy in air transport. The dynamic development of this industry has led to a change of perspective, according to which originally, until the 1970s, safety used to mean absence of accidents or of severe incidents. Since the mid-1990s, air safety management has become an organizational process, regarded as a core business function, and one for which staff members at the highest levels of the organization are responsible. Consequently, the sustainable development of this industry requires a combination of economic qualities and security. The "human factor" aspect, as an element with a significant impact on the level of safety,

is currently taken into account on every level of performance of tasks related to air transport. The best solution turns out to be introducing it at the earliest possible stage, to make sure threats can be predicted and eliminated efficiently. The conclusion drawn on the basis of experiences of the European Organisation for the Safety of Air Navigation "EUROCONTROL" (The Human Factors Case: Guidance for Human Factors Integration, 29.06.2007 (07/06/22–35) EUROCONTROL, https://www.skybrary.aero/bookshelf/books/4556.pdf (downloaded: 29.07.2020)) is that the sooner the "human factor" aspects are taken into account, the better the effect is later on. Consequently, an appropriate climate of action in the organization needs to be created to achieve a high level of safety. Organizations should be open and ready to share information on safety-related issues without fear that such information may be used against them and their employees. Therefore, taking actions for sustainable development also means caring for the quality of life of employees, not only in economic terms, but understood as the degree of satisfaction of people's needs related to their sense of safety at work, as the working environment is an important space for sustainable development on the individual level [33–37].

Limitations of the presented research certainly include the homogeneity of the research sample in cultural terms, as the respondents all came from a single country (Poland). An important direction highlighted in the literature on the subject, in the context of sustainable development, is to conduct comparative research on a wider scale, taking into account cultural differences. This requires an increase in the number of samples representing countries and all continents. As it is impossible to think about work safety culture without taking into account the cultural context, it is advisable, in relation to the aspect of sustainability, that comparative research should continue in different cultural contexts. Research is planned to be continued on representative groups of workers employed in the aviation industry in other countries. Undoubtedly, the occupational group of airport ground staff members is not homogeneous either in terms of the intensity of the impact of the various threats occurring in their working environment. Therefore, it is worth considering a comparative analysis of the results obtained, taking into account separate occupational subgroups.

We hope that the research we have carried out will contribute to the practical implications with regard to the sustainable development of the aviation industry, taking into account in particular the well-being of the people employed there, their sense of safety, and the maintenance of healthy working conditions.

**Author Contributions:** Conceptualization, M.D.; M.S.-S. and J.K.; methodology, M.D. and M.S.-S.; formal analysis, M.D.; M.S.-S. and J.K.; investigation, M.D.; M.S.-S. and J.K.; resources, M.D.; M.S.-S. and J.K.; data curation, M.D. and M.S.-S.; writing—original draft preparation, M.D.; M.S.-S. and J.K.; visualization, M.D.; M.S.-S. and J.K.; supervision, M.D.; M.S.-S. and J.K.; project administration, M.D.; M.S.-S. and J.K.; funding acquisition, M.D. All authors have read and agree to the published version of the manuscript.

**Funding:** The publication is financed within the framework of the programme titled "Dialogue" introduced by the Minister of Science and Higher Education between 2016–2019.

**Conflicts of Interest:** The authors declare no conflict of interest.

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
