# Peer review of "Safety Attitude as a Predictor of the Sense of Threat in the Workplace, Using the Example of Airport Ground Staff"

_sustainability, doi:10.3390/su12166569_

Round 1

Reviewer 1 Report

Citation musst be improved, there are also some spelling mistakes in it.

Analytical statistic results can be improve in presentation (better table results presentation)

Author Response

Thank you very much for the comments submitted. We have performed the following actions on their basis:

  1. Corrections of the text to remove errors.
  2. Preparation of tables in accordance with the APA system, correction of errors in Table 2.

Reviewer 2 Report

Dear Authors,

the topic of the study is very interesting and the study is well conducted. The presentation of the results and the discussion section have to be improved and in part rewritten. The specific aspects to be  revised are in the following suggestions.

In the abstract add information about some brief quantitative result of the study and some statistical data.

Line 36: Authors should redefine a shorter introduction.

Line 224: Has the study the approval of an Ethical Committee? If not the authors should explain the reasons

Line 228: How many individuals were contacted at the beginning of the study? What is the percentage of responders on the total of the subjects contacted?

Line 271: Put this phrase at the end of the Materials and Methods section in a paragraph with other following parts that describe the statistical analysis methods

Line 275-276: Put this phrase in Materials and Methods section in the final paragraph on statistical analysis methods

In table 2, what is the meaning of this four asteriscs after 0.31 in the second line of the table? It isn't explained in the notes

Line 325 and the following ones: Authors should rewrite the discussion and a brief section of conclusions. The discussion section should have the following structure: 1) the main results of the study; 2) Comparison of results with existing literature 3) limitations and strengths of the study

Kind regards

Author Response

Thank you very much for your interest in the subject of our research and for all the comments that helped us to improve the manuscript presented. We have sought to address every comment submitted. Below we explain the changes made.

  1. In the abstract, data has been introduced to make the results more specific in statistical terms.
  2. The introduction has been redrafted and shortened. The research group and the risks specific to it have been defined more precisely. Only the substantive scope of the variables discussed, namely the sense of threat and attitudes towards safety, has been addressed.
  3. The Ethics Committees on scientific research involving people at the Silesian University of Technology in Gliwice and at the University of Silesia in Katowice provide their opinion on scientific research projects that involve a risk to the physical or mental health of the participants, a privacy infringement risk, as well as potential other types of social or legal damage, taking into account scientific research projects using personal data. Our research did not involve such risks. All the respondents were adults who gave their voluntary and informed consent to participation in the study. The results obtained were coded with the anonymity of the participants ensured. We included this explanation in the manuscript.
  4. 299 individuals participated in the research, accounting for 91% of the people provided with the questionnaires. Information has been added in the paper.
  5. The study results were analyzed using the JASP 0.11.1 package (JASP Team, 2019) – the paragraph has been moved to the Materials and Methods section.
  6. An error has been corrected in Table 2 Relationships between safety attitude and sense of threat – correlation analysis.
  7. The discussion has been reworded in accordance with the structure suggested in the review, and limitations of the presented research have been indicated.

Reviewer 3 Report

make font consistent color

good tie to sustainability but would be enhanced with a reference or two....I think the topic also may also relate to the economic pillars given that personnel costs are a significant expense for all airports and airlines

page 2, paragraph lines 59-66 seems like a digression

Disagree with line 67 that many people see pilot as only person responsible for safety...this statement and much of the paragraph reduces credibility wrt familiarity of airport ops and airline ramp activities...

Would be good to reference any previous studies regarding safety culture and perceived threats/safety for airport workers

lines 87 thru 101...not all of these threats are equal and some do not typically apply to airport operations employees...tailoring the safety threats to more typical injuries and risks may make more sense (e.g., slips, trips and falls, jet blast, working around heavy and moving equipment, are a much greater risk than any of these dramatic examples)

line 191:  "airport ground staff members" would be helpful to clarify since this sounds like "airport operations workers" to me, whereas some previous references suggest or imply airline ramp workers -- big difference wrt duties, public vs. private employees, risks, etc.

Results

Provide more discussion and details about the statistical analysis. Discuss the implications of the data in the table.   Is M average and Me mean?  What does the magnitude of these numbers mean and what is the scale?  It's not clear how the likert scale answers from 1 to 5 translate to these numbers.

Since MDPI allows and supplemental files, consider including the data and questionaire.  If you publish your data it can be used and reference like a paper would be.

Keep all of Table 3 on the same page.

Don't overstate the findings since you have pretty low r and r2.  May be better to emphasize the value of the approach rather than the significance of the findings. 

Much of the lit review and discussion seems very theoretical without strong connections to the tight scope of the investigation.

Need to translate references to English.

None of the references address sustainability or aviation...seems like a gap:  include references for safety culture in aviation and work threats...this will allow you to address previous comments about understanding the most likely threats and being specific wrt the job description in commonly used aviation vernacular.

Author Response

Thank you very much for all the suggestions included in the review, helping us to improve the manuscript presented. We have made all efforts to address each of the comments. Below we present the changes made:

  1. In line with the valid point raised, the introduction has been redrafted, the group of airport ground staff members has been redefined, the presentation of the threats related to work at an airport has been narrowed down, and the assertions concerning the perception of the pilot’s role in creating a sense of safety for the users have been corrected.
  2. The information on the research tools making it possible to interpret the data in Table 2 has been supplemented.
  3. Reference has been made in the discussion to the importance of sustainable development in the creation of safe working environments that provide opportunities for mental well-being, with reference to the relationship between the research and the recommendations of the European Organisation for the Safety of Air Navigation. References to basic research in the field of social sciences on the sense of threat and the creation of attitudes conducive to safe behaviors have been used in the discussion, as there are no reports of similar explorations in the group of airport ground staff members. The authors continue to carry out international research in this area, consequently hoping that it will be possible to develop the results obtained in subsequent papers.

Round 2

Reviewer 2 Report

Dear authors,

all my requests were have been met.

Kind regards.

This manuscript is a resubmission of an earlier submission. The following is a list of the peer review reports and author responses from that submission.